# Camp as a Facilitator for Positive Childhood Experiences for Children and Youth with Serious Healthcare Needs: A Rapid Review

**DOI:** 10.3390/bs15111525

**Published:** 2025-11-10

**Authors:** Haley Pogachefsky, Ann Gillard, Laura Blaisdell, Christopher J. Stille, Robert Sege

**Affiliations:** 1Independent Researcher, Scarborough, ME 04074, USA; hpogachefsky@gmail.com; 2SeriousFun Children’s Network, Norwalk, CT 06855, USA; lblaisdell@seriousfun.org; 3Children’s Hospital Colorado, University of Colorado School of Medicine, Aurora, CO 80045, USA; christopher.stille@childrenscolorado.org; 4Tufts Medical Center, Boston, MA 02111, USA; robert.sege@tuftsmedicine.org

**Keywords:** medical specialty camp, positive childhood experiences, adverse childhood experiences, youth development, outdoors, children and youth with special health care needs

## Abstract

Children and youth with special healthcare needs (CYSHCN) face elevated risks of adverse childhood experiences while also having unique opportunities for positive childhood experiences (PCEs). Medical specialty camps can serve as protective environments promoting resilience and well-being in this population. We examined current literature to determine whether camp experiences align with the Healthy Outcomes from Positive Experience (HOPE) framework and function as PCEs for CYSHCN. A comprehensive literature search was conducted across PubMed, Google Scholar, and Elsevier databases using terms related to camps, positive childhood experiences, and childhood illness. Studies were systematically mapped onto the four HOPE framework categories and analyzed for qualities of effective PCE settings. Twenty-six studies demonstrated alignment between camp experiences and all four HOPE framework components: nurturing relationships, safe environments, social engagement opportunities, and social–emotional competency development. Four qualities of effective PCE settings emerged: being outdoors, engagement in meaningful activities, finding meaning in life, and experiencing “being away.” Research representing medical specialty camps demonstrates strong theoretical alignment with PCE frameworks, suggesting potential protective benefits against ACEs for CYSHCN. A conceptual model is proposed to guide future empirical research examining camps as facilitators of PCEs and their long-term health outcomes for this population.

## 1. Introduction

Children and youth with special healthcare needs (CYSHCNs) are those who have or are at risk for chronic physical, developmental, behavioral, or emotional conditions like asthma, autism, anxiety, epilepsy, or learning disorders, requiring specialized health and educational services to thrive, though each child’s needs are different ([30]). CYSHCNs comprised 20.8% of children in the United States in 2022–2023 ([10]). In addition to health diagnosis risks, these children are at increased risk of suicidal ideation, post-traumatic stress symptoms, lower socioeconomic attainment, higher risky health behaviors, and worse psychosocial functioning than their peers ([5]). CYSHCNs are also at higher risk of experiencing adverse childhood experiences than their healthy counterparts ([38]). Conversely, CYSHCNs also experience protective opportunities for healthy growth and development including access to quality healthcare, accessible environments, and family and community connections ([42]). Protected experiences like these are described in the Healthy Outcomes from Positive Experience (HOPE) framework as key positive childhood experiences (PCEs) that promote lifelong health and wellbeing in children and adolescents ([59]). Youth development settings such as camps have also been shown to lead to healthy growth and development of CYSHCNs (e.g., [48]). In this review, we set out to examine the fit between the HOPE framework and current literature about medical specialty camps to describe how they might produce PCEs that can function as a protective factor for CYSHCNs.

Medical specialty camps are specifically designed to serve CYSHCNs in outdoor settings. These camps consider multiple factors when determining appropriate programming: family and child interest in camp; a child’s health status, including mobility, mental health, and level of independence; access of children and their families to psychological, social, and structural supports; family and child ability to pay for the cost of camp and transportation; and the physical accessibility of the camp space. A positive camp experience is also founded on camps’ ability to provide accessible programming, medical support, appropriate dietary needs, and opportunities to sustain learnings from camp upon arrival home. For children for whom medical specialty camps are available, these experiences can be an impactful and lasting positive experience in their lives ([22]).

### 1.1. Adverse Childhood Experiences

Adverse Childhood Experiences (ACEs) are potentially traumatizing events that happen during childhood ([9]). ACEs include experiences of abuse, neglect, witnessing violence and substance abuse, and living in a home with a mentally ill parent ([9]). Experiencing one or more ACEs is directly correlated with negative health and social outcomes later in life ([9]), with approximately 64% of adults reporting having experienced at least one ACE ([38]). ACEs and the CYSHCNs population are closely linked. ACEs have also been associated with poor early childhood mental health and chronic medical conditions ([39]). The prevalence of ACEs is higher among CYSHCNs than among their peers without special health care needs (35% versus 17%) and CYSHCNs are more likely to experience two or more ACEs ([60]). More than a third of children with multiple ACEs (36.3%) have a special health care need, yet CYSHCNs are nearly four times as likely to have unmet health care needs compared to non-CYSHCNs ([33]). Further, in a population of CYSHCNs, [7] ([7]) found that children’s cancer symptoms can exacerbate pre-existing parental mental illness, which can, in turn, increase the risk of experiencing that particular ACE for the children themselves ([16]). Still, it is important to note that research on ACEs has not shown that medical trauma is an ACE for CYSHCNs, even though the experience of childhood illness seems to be a distinguishing ACE. These complex relationships between ACEs and childhood illness underscore the need for comprehensive, trauma-informed approaches to pediatric care.

However, the impact of ACEs on CYSHCNs is not uniform across all populations. The intersectionality of ACEs, racism, and CYSHCN status is of particular importance for Black CYSHCNs, who are more likely to experience a lack of culturally sensitive and family-oriented care and shortages of providers in their communities ([19]). Given the prevalence and close links between ACEs and serious illnesses in children and youth, more attention is needed to explore potentially mitigating factors.

### 1.2. Positive Childhood Experiences

Conversely, Positive Childhood Experiences (PCEs) are associated with improved health outcomes even in the presence of adverse childhood experiences. PCEs are characterized as a child’s “experience of having safe, stable, nurturing relationships and environments” ([57]). PCEs are experiences that protect against the adverse outcomes of adverse childhood experiences, and the more PCEs a child experiences, the more positive impact they receive ([3]; [12]; [43]). Studies show PCEs reduce the risk of depression, anxiety, and risky sexual behaviors, and increase positive body image, academic attainment, and mental health ([12]; [31]; [57]; [58]). Further, while not yet defined as a PCE, engagement with natural settings has been shown to support healing from adverse childhood experiences ([28]). Understanding camp’s benefits shows why more children with special healthcare needs should have access to these potentially healing experiences.

The Healthy Outcomes from Positive Experience (HOPE) framework defines PCEs and clear categories for understanding and implementing PCEs in various settings (Table 1). The HOPE framework places PCEs into four categories: (1) being in nurturing, supportive relationships, (2) living, developing, playing, and learning in safe, stable, protective, and equitable environments, (3) having opportunities for constructive social engagement and connectedness, and (4) learning social and emotional competencies ([59]). These categories are supported by several high-quality research studies (e.g., [57]; [58]). Supportive relationships with parents and caregivers contribute to secure attachment, setting the precedent for healthy relationships in middle childhood and beyond ([56]). Relationships with adults outside of the child’s family are also important, impacting the child’s self-regulation, sleep cycle, and psychosocial functioning in adulthood ([8]; [31]). Healthy, equitable environments can contribute to future stable housing, nutrition, sleep, and access to medical care in the future ([59]). Opportunities for social engagement that nurture a sense of belonging positively affect a child’s mental and emotional well-being in adolescence, and can reduce a child’s engagement in risky behaviors such as drinking, violence, and early initiation of sexual activities ([46]). Beyond exposure to social opportunities, explicitly learning social and emotional competencies can increase a child’s executive functioning and lay the groundwork for future physical health ([8]). While most PCE research has focused on the impact of nurturing, supportive relationships, a potential area of research is to better understand the impact of environment and social participation on PCEs ([31]).

Overnight camps are youth-centered settings that offer multiple PCEs from this framework. The outdoor recreation setting is a key feature of many camps. Outdoor recreation spaces provide a unique context where positive developmental outcomes among youth can be fostered ([32]; [1]). For example, camp participants’ “affinity for nature” increases with camp participation (e.g., [15]; [64]). Evolving research demonstrates that time in the outdoors can reduce stress by lowering heart rate and blood pressure, positively affecting cortisol levels, and improving cognitive functioning while indirectly increasing physical activity and social interactions ([40], [41]; [53], [54]). Such positive outcomes are particularly salient for CYSHCNs because they often spend a disproportionate amount of time in medical and other indoor settings compared to healthier counterparts. However, more research is needed to identify which specific types of programs can improve well-being and to understand process mechanisms ([18]). It is not known whether camp participation could facilitate protective PCEs against the adverse childhood experiences associated with childhood illness. Research is needed to elucidate the means and outcomes that camp-based settings could play in improving the well-being of CYSHCNs through PCEs.

### 1.3. Purpose

This rapid review had two aims. First, we aimed to explore whether and how camp, as an outdoor-centered environment that fosters connection to the outdoors through activities like hiking, swimming, and outdoor living, can be a protective factor for CYSHCNs ([59]). We explored this first aim through a literature review with two guiding questions: (1) Using the HOPE framework, what experiences within camp serve as PCEs? and (2) What are the qualities of effective PCE settings as they relate to camp and CYSHCNs?

A second aim of this rapid review was to create a conceptual model to articulate the mechanisms for known psychosocial and health-related outcomes of outdoor-based camp experiences. Camp environments typically provide rich opportunities for outdoor-based activities, outdoor skill development, and environmental appreciation that can uniquely contribute to positive developmental outcomes. This conceptual model could be used for future studies, specifically exploration into the long-term health outcomes of CYSHCNs who engage in outdoor recreation such as camp, compared to their non-camper counterparts. These distal outcomes closely mirror the health outcomes of PCEs, indicating a potential area for future understanding of camp as a PCE.

## 2. Methods

A comprehensive search was performed including PubMed, Google Scholar, and Elsevier databases using search terms “camp AND positive childhood experience.” Because no studies have examined this specific relationship before, searches were broadened to terms, “Childhood illness AND positive childhood experience,” “Counter-ACE,” “Occupation AND health,” “Ecotherapy,” “Positive youth development,” “Childhood cancer AND trauma.” The review was extended through the reference lists of articles found in the database searches. The [35] ([35]) critical appraisal checklist for systematic reviews was utilized and all criteria met. Searches were conducted separately by the first two authors, who collaboratively assessed articles for relevance to the study, coming to consensus regarding inclusion or exclusion of articles. The full research team reviewed the theoretical model and manuscript to further substantiate inclusion of each article. Article abstracts were reviewed and studies reviewed in full if they met the following criteria: (1) available in English, and (2) peer reviewed. Full studies that met the above criteria were further reviewed, and were included if they met a third criteria, (3) describing both qualities of camp as a PCE and qualities of an effective PCE setting. Thirty-eight studies were excluded that did not describe both qualities of camp as a PCE and qualities of an effective PCE setting. Using inductive reasoning, the 26 included studies were counted into in two sets of groups to address the two aims of the present study. First, studies were mapped onto the four qualities of PCEs according to the HOPE framework to explore the relationship between the qualities of camp and those of a PCE. Second, studies were grouped according to key qualities of effective PCE settings as they were derived via inductive reasoning, to demonstrate camp as a high-quality setting for children to experience PCEs. A conceptual model was then derived through review and discussion amongst the authors to model the associations between camp and PCEs.

## 3. Results

No specific studies of camp as a facilitator of PCEs were found, indicating a rich area for future research. We identified twenty-six studies (Figure 1) linking PCE qualities described in the HOPE framework to corresponding qualities found in camps, which in turn revealed four qualities of effective PCE settings (Table 2). Thirteen of the twenty-six studies were conducted in the United States, and thirteen were conducted abroad in countries including Sweden, Romania, China, Australia, and Greece. In the following sections, we will examine specific studies that investigated each of these four identified qualities.

## 4. Qualities of PCEs in the HOPE Framework

### 4.1. Being in Nurturing, Supportive Relationships

Meaningful relationships can serve as a cornerstone of well-being for CYSHCNs facing health challenges. [27] ([27]) found respondents recovering after life threatening illness focused less on professional healthcare resources than relationships and informal care received. Relational care is abundant in camp experiences; in their systematic review of camp literature [63] ([63]) found that all camps, regardless of specialty, utilize strategies to foster nurturing and supportive relationships between campers and adults, creating an environment that prioritizes and champions relationships. Furthermore, engagement in medical specialty camps can improve relationships between CYSHCNs and their medical providers upon return home by creating therapeutic alliance through shared experiences ([13]).

### 4.2. Living, Developing, Playing, and Learning in Safe, Stable, Protective, and Equitable Environments

The natural setting of camps provides an opportunity for CYSHCNs to experience safety, stability, protectiveness, and equity outside of the hospital or clinic setting. [50] ([50]) identified that natural environments create emotional safety for campers. Many camps provide opportunities for campers to connect and build a relationship to the outdoors, even if being outdoors is not a key focus of the camp experience ([11]). Camp also represents a respite from stressors that can be inherent to campers’ home lives, which can provide experiences of physical and emotional safety ([7]; [50]). [52] ([52]) also found that attendance at camp can increase participation in medical routines and healthcare for campers with serious healthcare needs, indicating an environment supportive and equitable enough to afford increased engagement. Camp provides opportunities for novel engagement in daily activities and life-skill building, including leisure skill building ([65]). Many camps encourage “challenge by choice,” a phrase representing the effort to create the right environment for safe risk taking tailored to each child’s level of comfort ([23]).

### 4.3. Having Opportunities for Constructive Social Engagement and Connectedness

Camp experiences are designed to provide positive social engagement by incorporating program elements championing peer-peer and camper-counselor relationships that can have a lasting impact on campers’ lives ([63]). Indeed, two of the six C’s of positive youth development (Connection, Confidence, Caring, Competence, Character, Contribution), a model followed by many camps, are Connection and Caring ([20]; [2]). The meaningful connections made at camp are a logical and likely inherent part of the camp experience. These relationships are largely positive and can facilitate the development of friendship skills among campers ([22]).

### 4.4. Learning Social and Emotional Competencies

However, relationships at camp are not ubiquitously positive. As in any community, challenges can also be part of the camp experience. Intentional management of moments of tension and friction contribute to positive youth development by providing opportunities for agency, self-regulation, and character development ([20]). In the best cases, these interactions are moderated by knowledgeable and sensitive staff trained in social–emotional skills themselves ([61]). Camp experiences can provide a supportive environment, combined with scaffolded skill building, for campers to learn social and emotional competencies through trial and error, and direct instruction from staff.

### 4.5. Summary

Camps prioritize nurturing relationships between campers and adults while providing natural, safe environments that offer respite from medical settings and home stressors. These settings can facilitate constructive social engagement through intentionally designed peer and counselor relationships that support positive youth development. Additionally, camps create supportive contexts for developing social and emotional competencies, where trained staff help campers navigate both positive interactions and challenging moments to build essential life skills.

## 5. Qualities of PCE Settings

In this rapid review, a pattern emerged between effective settings generally, and camp settings in particular. The literature on therapeutic PCE settings was grouped into four broad categories for this rapid review: (1) being outdoors, (2) engagement in meaningful activities, (3) finding meaning in life and (4) having a feeling of “being away.” All four categories map onto common qualities of the camp setting, indicating a connection between camp experiences and experiences generally considered to be PCEs.

### 5.1. Being Outdoors

Being outdoors can create an emotionally safe space for connection to other people who share a similar diagnosis, increasing social cohesion and showing participants that “they are not alone with their disorder” ([36]). Time in the outdoors has been shown to increase oxytocin levels, inhibiting activation of the stress response system ([25]; [47]), while living in a green space is correlated with higher self-reported indicators of both physical and mental health ([45]). Being outdoors can increase a sense of safety, a crucial component of childhood PCEs ([49]). The ECO-Therapy Model developed by [17] ([17]) posits that nature-based therapy sessions have the capacity to build self-agency and adaptive capacity. Camp provides ample opportunity for outdoor exploration, leisure, and learning, and has been shown to increase children’s appreciation of and positive actions towards nature ([11]; [22]).

### 5.2. Engagement in Meaningful Activities

Engaging in meaningful activities can be both a means and an end to wellness and health ([26]). [24] ([24]) describes that the meaning of an activity relates to “the activity’s congruity with one’s value system and needs, its ability to provide evidence of competence and mastery and its value in one’s social and cultural group.” [44] ([44]) found six outcomes associated with engaging in meaningful activities with peers who shared a medical diagnosis: (1) making meaning, (2) expressing thoughts and emotions, (3) changing physical, emotional, and cognitive states (4) cultivating skills, strengths, and virtues; (5) connecting and belonging, and (6) making a contribution ([44]). Meaningful activities at camp for CYSHCNs can range from participation in activities of daily living (ADLs) like getting dressed and brushing teeth with increased independence ([29]; [34]), to storytelling ([14]), to leisure activities like gardening ([49]) and crafts ([62]).

### 5.3. Finding Meaning in Life

Meaning in life is defined as the feeling that an individual’s existence has significance, purpose, and coherence ([55]). Finding meaning in life can contribute to adolescents’ psychological health ([34]) and protect against health risk behaviors like drug abuse, unsafe sex, and lack of exercise ([4]). The American Academy of Pediatrics’ blueprint for change identifies sense of self-worth and purpose as critical components to promote and support flourishing ([6]).

Meaning can be made through engagement in everyday activities ([44]; [29]) For example, [49] ([49]) found that participation in leisure activities is positively correlated with self-efficacy and self-worth. These positive elements of self-concept reflect meaning in life because when people believe their lives matter, they are more likely to engage in behaviors that support their health and thriving ([55]). Camp offers opportunities for development of leisure skills that are transferable to other settings in a child’s life ([65]), indicating opportunities for camp attendees to derive meaning not only in the camp context, but also back home with access to the same leisure activities. Camps for CYSHCNs have been shown to facilitate identity development, confidence, and belonging ([22]), which are key ingredients for finding meaning in life.

### 5.4. Having a Feeling of “Being Away”

[51] ([51]) identified that a key component of a restorative therapeutic environment involves creating a sense of “being away.” This aligns with [37] ([37]) framework of the restorative benefits of the outdoors. Our rapid review found that camp can provide an opportunity for a feeling of being away by presenting campers with novel activities, relationships, and exposure to the outdoors ([11]; [50]). Camp can foster both the sense of being away as a mental and a physical state. The many different experiences of emotions while away at camp can provide learning opportunities for intentional self-regulation ([21]; [20]). Camp can also provide opportunities for novel leisure activities campers would not typically participate in at home such as crafts and storytelling, exploration of which can provide an additional sense of being away ([14]; [62]).

### 5.5. Summary

Four categories of effective PCE settings aligned with camp experiences: being outdoors, engagement in meaningful activities, finding meaning in life, and having a feeling of “being away.” Outdoor environments can improve safety, social cohesion, and stress while meaningful activities at camp—from daily living skills to creative pursuits—promote skill development, emotional expression, and peer connection. Camp experiences can facilitate meaning-making through identity development and belonging, which can serve as protective factors for CYSHCNs. Additionally, camps can provide the restorative “being away” experience through novel activities and natural settings, offering both physical and mental respite from the home environment.

## 6. Discussion

The purpose of this rapid review was to explore the qualities of camp as a facilitator of PCEs using the HOPE framework and the characteristics of settings specifically related to camp programs for CYSHCNs. While we did not find existing research connecting camps as a PCE for CYSHCNs, we did find good alignment between the identified studies and the HOPE framework. We also found four qualities of effective PCE settings that closely aligned with existing research on medical specialty camps for CYSHCNs. Importantly for this special issue, we identified key studies linking the outdoor setting of camp to PCEs and qualities of PCE settings: being outdoors, meaningful activities, finding meaning in life, and having a feeling of being away. This research also contributes to the existing literature of promising practices to support CYSHCNs ([59]) and advances the American Academy of Pediatrics’ blueprint for change to improve the lives and well-being of CYSHCNs ([6]).

There was significant overlap between the qualities outlined in the HOPE framework and the qualities of a camp setting, particularly the restorative benefits of natural environments. This overlap provides a conceptual basis to understand how camp can fit the description of a PCE, though future research is needed to understand whether camp can serve as a protective factor against the adverse childhood experiences associated with childhood illness. To meet our second study aim, next we discuss our conceptual model.

## 7. Conceptual Model

Based on results from the rapid review and the authors’ professional expertise, we created a conceptual model describing the known and expected outcomes of camp, and the contexts where CYSHCNs have access to a high-quality camp experience that leverages outdoor environments for healing and growth (Figure 2). Specifically, our rapid review contributed to the “Characteristics of Camp as a Positive Childhood Experience” and “Short Term Outcomes” sections.

This framework can be used to conceptualize future studies, specifically research into the distal (medium- and long-term) health outcomes of campers with special healthcare needs, compared to their non-camper counterparts. These distal outcomes closely mirror health-related outcomes of PCEs, indicating a potential area for future understanding of medical specialty camp as a PCE. Providing a research foundation that connects CYSHCNs with camp and with PCEs is important for linking these disparate fields, and ultimately creating conditions where CYSHCNs can thrive.

## 8. Limitations

This rapid review has several important limitations. First, we did not find research that directly examined medical specialty camps as facilitators of positive childhood experiences for CYSHCNs, so we synthesized findings across disparate fields to establish connections. Second, the review was limited to English-language, peer-reviewed publications, which could have excluded relevant studies in other languages or grey literature. Finally, the conceptual model proposed requires empirical validation through additional research to confirm whether camp participation actually leads to the hypothesized protective factors of PCEs in this population.

## 9. Implications for Future Research

Because no existing studies investigate medical specialty camps as a PCE for CYSHCNs, future research is needed to understand whether a camp experience can serve as a protective factor against adverse childhood experiences associated with life-threatening illness. Our study found limitations in the existing evidence base, particularly regarding study quality and generalizability across cultural contexts, indicate the need for future research. Additional research is needed to understand the health outcomes associated with camp attendance, including potential mechanisms behind improved mental and behavioral health conditions. Examining the role of outdoor experiences in promoting mental health is of particular importance. Long-term outcomes in CYSHCNs, such as thriving, meaning and purpose, social connectedness, quality of life, and health metrics, are further areas of future study.

## 10. Implications for Practice and Policy

Understanding the aspects of camp that map onto the qualities of PCEs can inform best practices at camps serving CYSHCNs. Camps have a long history of intentional programming, and designing activities and settings to ensure the camp experience is maximally aligned to serve as a PCE for all children, and especially CYSHCNs, is a potential important practice change. The HOPE framework provides guidance to map existing activities and structures to outdoor settings, meaningful activities, meaning-finding opportunities, and feelings of being away. For example, camps could consider prioritizing meaningful relationship skill-building between campers with similar health experiences, thereby increasing a sense of belonging and peer-peer connections within the therapeutic context of natural settings. Camps might also consider helping campers more explicitly reflect on the meaning of their camp experiences and their connection with outdoor activities, and how these experiences can be brought back out into their non-camp lives. With intentional PCE practices, camps can maximize the positive health outcomes for campers and potentially mitigate the impacts of adverse childhood experiences associated with life-threatening childhood illness. Staff training could involve learning about the HOPE framework and PCE principles, equipping counselors to recognize and facilitate moments of meaning-making and connection, and developing competencies in supporting peer relationships among CYSHCNs. Camps can also maximize the restorative benefits of natural environments by designing activities that encourage meaningful engagement with nature rather than passive presence outdoors. Further, evidence that camps can serve as a protective factor can validate and encourage funding and support for such interventions so that more CYSHCNs can attend and receive the benefits of a high-quality camp experience. Policymakers could integrate camp into comprehensive care models for CYSHCNs by recognizing medical specialty camps as therapeutic interventions within healthcare systems, and developing policies that facilitate collaboration between medical providers and camp programs.

## 11. Conclusions

In this paper we examined the existing literature investigating how camp qualities align with the HOPE framework of PCEs. Camp provides opportunities to experience the four components of the HOPE model: relationships, environment, social connectedness, and social–emotional competencies. Within each of these categories, evidence suggests that qualities of camps for CYSHCNs align with the HOPE model, and thus camp could be a protective factor against adverse childhood experiences for CYSHCNs.

Furthermore, four categories of effective PCEs were elucidated from the literature: being outdoors, engaging in occupations, finding meaning in life, and a feeling of “being away.” These categories help define the camp experience and support our conceptual framework, which positions camps as an effective form of PCEs. Finally, we suggested areas for future research to examine whether camp may be considered a PCE for CYSHCNs to protect against the adverse childhood experiences associated with childhood illness.

## Figures and Tables

**Figure 1 behavsci-15-01525-f001:**
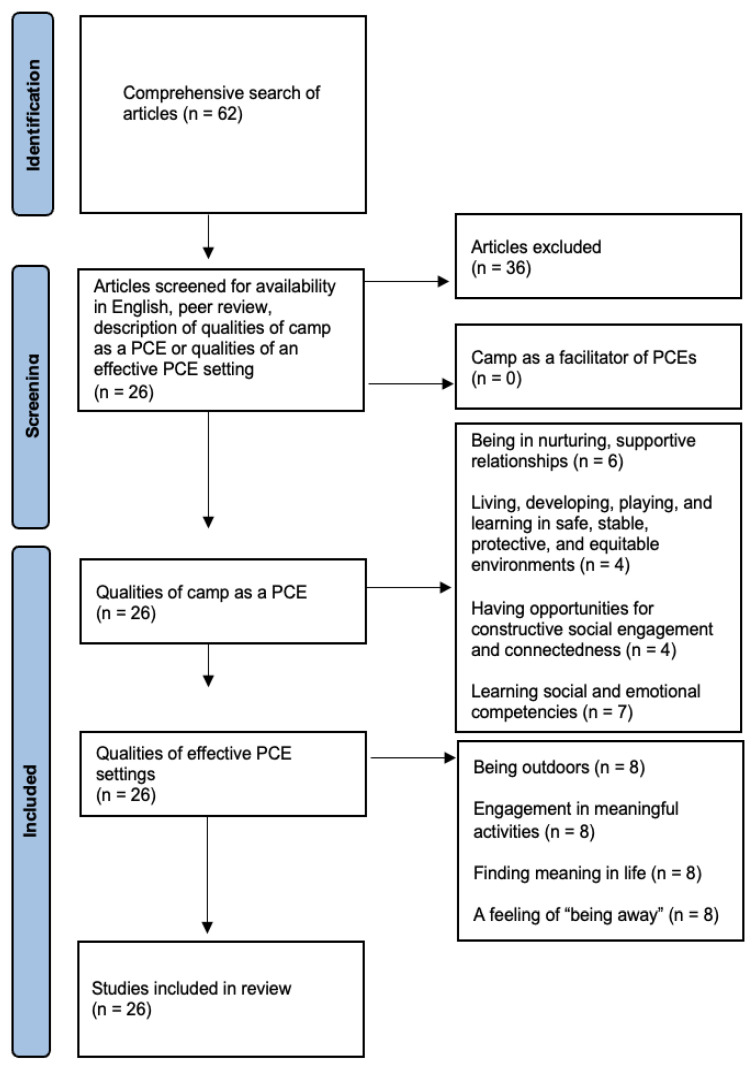
Rapid Review Process.

**Figure 2 behavsci-15-01525-f002:**
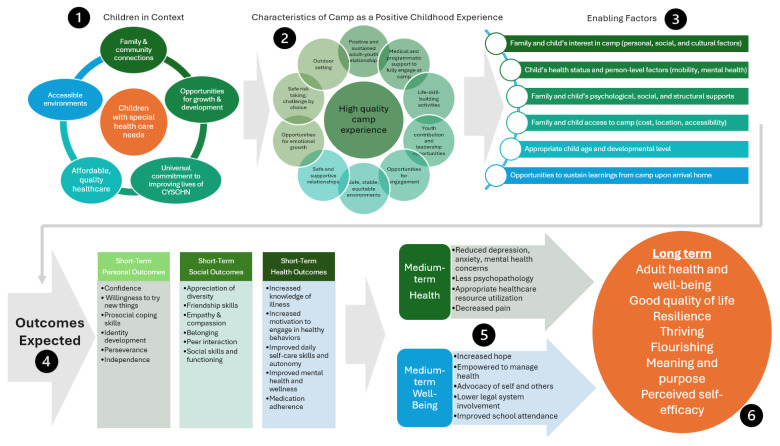
Conceptual Model of Camps as Positive Childhood Experiences for Children and Youth with Special Health Care Needs.

**Table 1 behavsci-15-01525-t001:** Healthy Outcomes from Positive Experiences Framework ([59]).

Outcome	Definition
*Relationships*	Being in nurturing, supportive relationships
*Environment*	Living, developing, playing, and learning in safe, stable, protective, and equitable environments
*Engagement*	Having opportunities for constructive social engagement and connectedness
*Emotional Growth*	Learning social and emotional competencies

**Table 2 behavsci-15-01525-t002:** Rapid Review Results.

Comprehensive Search Included in This Study	n = 62 n = 26		
Qualities of camp as a positive childhood experience (PCE)	Being in nurturing, supportive relationships	[22] ([22]); [27] ([27]); [36] ([36]); [63] ([63]); [6] ([6]); [13] ([13])	n = 6
	Living, developing, playing, and learning in safe, stable, protective, and equitable environments	[43] ([43]); [22] ([22]); [63] ([63]); [11] ([11])	n = 4
	Having opportunities for constructive social engagement and connectedness	[36] ([36]); [63] ([63]); [2] ([2]); [22] ([22])	n = 4
	Learning social and emotional competencies	[3] ([3]); [22] ([22]); [36] ([36]); [63] ([63]); [20] ([20]); [2] ([2]); [61] ([61])	n = 7
Qualities of effective PCE settings	Being outdoors	[36] ([36]); [50] ([50]); [25] ([25]); [45] ([45]); [11] ([11]); [49] ([49]); [17] ([17]); [22] ([22])	n = 8
	Engagement in meaningful activities	[52] ([52]); [22] ([22]); [44] ([44]); [29] ([29]); [34] ([34]); [62] ([62]); [49] ([49]); [14] ([14])	n = 8
	Finding meaning in life	[52] ([52]); [22] ([22]); [44] ([44]); [4] ([4]); [6] ([6]); [49] ([49]); [55] ([55]); [65] ([65])	n = 8
	A feeling of “being away”	[50] ([50], [51]); [20] ([20]); [2] ([2]); [11] ([11]); [21] ([21]); [14] ([14]); [62] ([62])	n = 8

## Data Availability

The original contributions presented in this study are included in the article. Further inquiries can be directed to the corresponding author.

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
