# Peer review of "Camp as a Facilitator for Positive Childhood Experiences for Children and Youth with Serious Healthcare Needs: A Rapid Review"

_behavsci, 2025, doi:10.3390/bs15111525_

Round 1

Reviewer 1 Report

Comments and Suggestions for Authors

Thank you for your detailed work in this under-developed area of intervention for children and youth with health needs. The notion of camps and outdoor adventure experiences is of increasing interest internationally. It would be of benefit to identify or define what you mean by "Special Healthcare Needs" for example does this include developmental learning needs such as Dyslexia, Developmental Language disorder and intellectual disability?

The writing is clear and grammatically correct, but the extensive use of acronyms, although described, makes it hard to read. Sometimes the language is overly technical or inaccurate e.g. "Understanding and quantifying the potential positive impact of camp provides a rationale for improved access and engagement in these protective and potentially healing experiences for CYSHCN." If the intention is to determine whether there is a positive impact, then you might want to reword this aim - you might not want improved access if it doesn't help.

The method does not indicate that you counted the studies into groups, yet you have reported that. What is the significance of the counting? How does counting the articles relate to your research aims?

Figure 1. The columns would benefit from headings.

I'm not clear why 38 articles were excluded - it would be useful to make this more obvious. Are the side boxes articles included or excluded?

Figure 3. You say relates the articles mapped to four qualities, but there are 8 subcategories and two categories so I find this quite confusing. I think this table could be much better laid out to make use of space. It would be useful to indicate in the reference list or otherwise, which were the included articles.

The conceptual model does appear to capture many of the factors from the literature but I am not clear how they relate to each other. I think some direction or structure between the top row and the bottom row of the overarching picture would help. I note that there is no reference to the outdoors or nature in the model which seems to be of utmost importance. A conceptual map as a guide for future research is a valuable contribution to this area of study. Perhaps the counted articles could be linked to the conceptual map as an indicator of gaps in our knowledge to date.

I wish you well.

Comments on the Quality of English Language

As before. I have no concerns in general about the standard of English.

Author Response

Please see attached file for list of changes made. The revised manuscript has track changes included.

Reviewer 2 Report

Comments and Suggestions for Authors

The manuscript entitled “Camp as a facilitator for positive childhood experience for children and youth with serious healthcare needs: A rapid review” addresses an important and timely topic. The authors successfully highlight how medical specialty camps may align with the HOPE framework and function as positive childhood experiences for children and youth with special healthcare needs (CYSHCN). The integration of existing literature into the categories of nurturing relationships, safe environments, social engagement, and socio-emotional learning is a strength, and the conceptual model proposed is a valuable contribution that could guide future empirical work.

The review is well-structured and clearly written, and the rationale for focusing on camps as potentially protective environments is compelling. The discussion appropriately situates the findings within broader literature on ACEs and PCEs, underscoring the need for further exploration of long-term outcomes.

That said, there are areas where the manuscript could be further strengthened. First, the methods section would benefit from more detail on search strategies and inclusion/exclusion criteria to enhance transparency and reproducibility. Second, while the discussion is thorough, it may be useful to provide a more critical appraisal of the limitations of the existing evidence base, particularly regarding study quality and generalizability across cultural contexts. Finally, more explicit recommendations for camp practitioners and policymakers, beyond future research directions, could increase the paper’s applied value.

Overall, this manuscript is a thoughtful and relevant contribution to the literature on CYSHCN and positive youth development, and with some refinements, it has strong potential for publication

Author Response

(The authors gave the same response as above.)

Reviewer 3 Report

Comments and Suggestions for Authors

Thank you for the opportunity to review this manuscript. Discussing camp for CYSHCN as a potential Protective Childhood Experience is an interesting argument and worth consideration. Overall, the paper was well written, but it needs more detail, specifically in the methods, to clarify the exact procedures. My specific comments are below:

  1. On page 3, lines 99-102 are hard to follow. I suggest revising these sentences for clarity.
  2. Much more detail is needed about the methodology.
    1. Who conducted, and what criteria were used during the screening and full text review phases of the review?
    2. How was goodness of fit with the 2 study goals determined for the articles reviewed? By how/how many reviewers? How were disagreements resolved?
    3. How were the settings for PCEs derived? How were articles used to develop these settings and/or how were articles included/excluded from these categories?
  3. On page 6, lines 205-206 – how/why can MSCs improve relationships between CYSHCN 205 and their medical providers upon return home?
  4. On page 6, lines 250-252, how were the categories determined? How were studies grouped into categories or to determine categories?
  5. Page 9, limitations section – the second sentence needs to be revised or split into at least 2 sentences for clarity.
  6. No limitations are discussed.

Author Response

(The authors gave the same response as above.)
